# Effect of Wind-Induced Vibration on Measurement Range of Microcantiflever Anemometer

**DOI:** 10.3390/mi13050720

**Published:** 2022-04-30

**Authors:** Yizhou Ye, Shu Wan, Xuefeng He

**Affiliations:** Key Laboratory of Optoelectronic Technology & Systems, Ministry of Education, Chongqing University, Chongqing 400044, China; wanshu@cqu.edu.cn (S.W.); hexuefeng@cqu.edu.cn (X.H.)

**Keywords:** anemometer, flow sensor, wind-induced vibration, microcantilever

## Abstract

In this paper, the effect of wind-induced vibration on measurement range of microcantilever anemometer is investigated for the first time. The microcantilever anemometer is composed of a flexible substrate and a piezoresistor. The wind speed can be detected through the airflow-induced deformation in the flexible substrate. Previous work indicated that the flexible substrate vibrates violently once the wind speed exceeds a critical value, resulting in severe output jitter. This wind-induced vibration limits the measurement range of the anemometer, and the relationship between the anemometer measurement range and its structural parameters has not been explored systematically. Therefore, this paper aims to reveal this relationship theoretically and experimentally, demonstrating that a shorter and thicker cantilever with larger stiffness can effectively suppress the wind-induced vibration, leading to the critical speed rising. By eliminating the wind-induced vibration, the measurement range of the microcantilever anemometer can be increased by up to 697%. These results presented in this paper can pave the way for the design and fabrication of wide-range mechanical anemometers.

## 1. Introduction

The wind-induced vibration is a physical phenomenon that exists widely in nature. In the field of energy harvesting, wind-induced vibration has been extensively utilized to convert the ambient fluid flows into electricity [1,2,3]. The generated electricity can power wireless sensor nodes without batteries [4,5]. P. Chen et al. proposed a self-power management system that can realize automatic irrigation, weather monitoring, and wireless water level warning by harvesting wind energy [6]. X. Yang et al. presented a piezoelectric wind energy harvester enhanced by the interaction between vortex-induced vibration and galloping to power wireless temperature and humidity sensing nodes [7]. In these self-power systems, stronger wind-induced vibration is desired to create more energy. Hence, the research objective is focused on vibration enhancement. However, wind-induced vibration is not always desirable. In the afternoon of 5 May 2020, the Humen Bridge of China suffered a sudden wind-induced vibration, resulting in serious traffic jams and arousing strong public attention [8]. In addition, in previous work, our group has fabricated a microcantilever anemometer, as shown in Figure 1 [9]. At low wind speeds, the microcantilever undergoes stable deformation, and the wind speed can be determined by detecting the deflection of the cantilever. However, once the wind speed exceeds a critical value, the microcantilever starts to shake, resulting in severe jitter of the anemometer output. The winds higher than the critical speed become impossible to be measured. In this experiment, the wind-induced vibration limits the ability of the anemometer to monitor high winds. Therefore, the suppression of wind-induced vibration is also a significantly important research field.

To date, almost all of the wind-induced vibration suppression research is focused on the fields of civil and aerospace engineering [10,11,12,13,14]. Most research objects are large-scale systems, such as buildings, bridges, aircraft, pipelines, and transmission lines. There is still no systematic study on the wind-induced vibration suppression of micro-scale anemometers. Therefore, this paper discusses the wind-induced vibration theory about the microcantilever anemometers, and proposes the corresponding methods to suppress this vibration. On this basis, anemometers prototypes with different structural parameters are designed and fabricated to examine the theory. The proposal of the vibration elimination methods provides a vital reference for improving the measurement range of the microcantilever anemometers.

## 2. Principle and Design

The microcantilever anemometer used to wind speed measurement is composed of a flexible substrate and a copper constantan piezoresistor, as shown in Figure 2. At low wind speeds, the flexible substrate deforms steadily with the wind, and there is a correspondence between the wind speed and the substrate deflection. In contrast, the flexible substrate will undergo wind-induced vibration when the wind speed is close to or exceeding the critical value. This vibration significantly affects the output signal of the anemometer, making the wind speed measurement unreliable. In this device, the piezoresistor is used to detect the deformation or vibration behavior of the microcantilever anemometer.

For low wind speeds (*U* < *U*_v_), the deflection *w*(*x*) of the flexible substrate can be expressed by [15]
(1)$$w(x)=\frac{px^{2}}{24EI}(x^{2}-4Lx+6L^{2})$$
where *E* and *I* are the elasticity modulus and the area inertia moment of the cantilever, respectively. *L* is the length of the movable flexible substrate, and *x* is the coordinate in Figure 2a. Additionally, *p* is the distributed load exerted by the wind, which can be expressed as follows [16]
(2)$$p=\frac{1}{2}C_{D}\rho_{air}WU^{2}$$
where *C_D_*, *ρ_air_*, *W* are the coefficient of drag force, the air density, and the cantilever width, respectively. Hence, the deflection *w*(*x*) of the flexible substrate can be given by
(3)$$w(x)=\frac{C_{D}\rho_{air}Wx^{2}(x^{2}-4Lx+6L^{2})}{48EI}U^{2}$$

The deflection *w*(*x*) of the flexible substrate is proportional to the square of wind speed *U*. This is consistent with the results described by the Bernoulli equation [16].

However, when the wind speed exceeds the critical value (*U* > *U*_v_), the airflow-induced vibration will happen. As a result, the Bernoulli equation becomes useless to predict the motion of the cantilever. According to the theory of continuous system vibration, the motion equation for the transverse vibration of the cantilever can be given by [15]
(4)$$EI\frac{\partial^{4}w(x,t)}{\partial x^{4}}+\rho_{can}A\frac{\partial^{2}w(x,t)}{\partial t^{2}}=p$$
where *w*(*x*,*t*), *ρ_can_*, and *A* are the time-dependent transverse deflection, the cantilever density, and the cross-sectional area of the cantilever, respectively. For microcantilevers, it is vortex-induced vibration that causes the shake of the output [17]. When an airflow acts on the cantilever, periodic shedding of vortices will appear. The corresponding vortex shedding frequency *f_v_* is expressed by [17,18]
(5)$$f_{v}=S_{t}\frac{U}{D}$$
where *S_t_* is the Strouhal number, which depends on the shape of the cantilever and is typically 0.15–0.25 for an arbitrary shape. *D* is the windward-side characteristic size of the cantilever. The shedding of vortices causes a fluid oscillation force with frequency *f_v_*, resulting in the cantilever vibration. When the trigger frequency *f_v_* is close to the natural frequency *f_o_* of the cantilever, the vortex-induced vibration will occur, which means
(6)$$f_{v}=f_{o}$$
where the natural frequency *f_o_* of the cantilever can be written as [15,19]
(7)$$f_{o}=\frac{1}{2\pi }{\left(\beta_{n}l\right)}^{2}\sqrt{\frac{EI}{\rho_{can}AL^{4}}}$$
where *β_n_l* is constant, which is expressed by
(8)$$\beta_{n}l\approx (2n-1)\pi /2$$

Therefore, the vortex-induced vibration critical speed *U*_v_ of the cantilever can be written as
(9)$$U_{v}=f_{o}\frac{D}{S_{t}}=\frac{1}{2\pi }{\left(\beta_{n}l\right)}^{2}\sqrt{\frac{EI}{\rho_{can}AL^{4}}}\frac{D}{S_{t}}=\frac{{\left(\beta_{n}l\right)}^{2}}{4\pi S_{t}}\frac{T}{L}\sqrt{\frac{E}{3\rho_{can}}}$$
where *T* is the thickness of the cantilever. It is evident that the critical wind speed *U_v_* is proportional to *T* and *E*^1/2^, and inverse proportional to *L*. Therefore, to improve the critical speed, one can then look into three directions: increasing cantilever thickness *T*, increasing elasticity modulus *E*, and reducing cantilever length *L*.

## 3. Experiment and Discussion

Given the analyses above, 12 microcantilever anemometer prototypes with structural parameters in Table 1 were fabricated and tested in the wind tunnel to validate the deduced results. For prototypes 1 to 9, the cantilevers were formed with PET (polyethylene terephthalate) substrate, while prototypes 10, 11, and 12 were made of PC (polycarbonate) and two different PVC (polyvinyl chloride) materials. The piezoresistor unit fabricated by MEMS (Micro-Electro-Mechanical Systems) technology was pasted onto the root of the flexible substrate to measure the cantilever deformation, as shown in Figure 3. Owing to the ultrathin thickness and low Young’s modulus, the effect of the piezoresistor unit substrate on the cantilever deflection can be omitted. In the measurement, the piezoresistor was inserted in a Wheatstone bridge, and an instrument amplifier AD623 was utilized to amplify the bridge output signal. A high-precision multimeter received the output signal of the amplifier and recorded the deformation of the cantilever. When the jitter range of the output voltage is more extensive than 10 mV, the wind-induced vibration is considered to occur. The overall experiment setup for the cantilever anemometer is shown in Figure 3.

Prototypes 1 to 5 were firstly tested to investigate the effect of the cantilever length. The width and thickness of the cantilevers are set to be 15 mm and 0.2 mm, respectively, and the length changes from 20 mm to 40 mm with an interval of 5 mm. The voltage outputs of these prototypes at different wind speeds are recorded in Figure 4. It can be seen that for a single cantilever, the deflection increases with the wind speed. As for different cantilevers, the deflection increases with the length. These results are consistent with the outcome predicted by Equation (3). In addition, once the wind speed is higher than the critical value, the voltage outputs of the cantilever anemometers will vibrate violently. When the fluctuation amplitude of the output voltages exceeds 10 mV, it can be considered that the wind-induced vibration happens. The corresponding wind speed is the critical value. The inset in Figure 4 gives the relationship between the critical wind speed and the cantilever lengths. It can be observed that the critical speed becomes higher with the length reduction. This result reveals that shortening the cantilever length can suppress the wind-induced vibration and improve the critical wind speed, corresponding with the predicted impacts of Equation (9). Furthermore, for the anemometer, the measurement sensitivity is defined as the slope of the output signal curve [20]. Therefore, the results also show that the cantilever anemometer presents higher measurement sensitivity in high winds, and higher critical speed is accompanied by lower measurement sensitivity.

Prototypes 3, 6 to 9 were also characterized to explore the effect of the cantilever thickness. The length and width of the cantilevers are set to be 30 mm and 15 mm. When different thicknesses (0.2 mm, 0.3 mm, 0.4 mm, 0.5 mm, 0.6 mm) are chosen, the voltage outputs corresponding to different wind speeds are presented in Figure 5. It can be observed that a thinner cantilever results in more vigorous deflection and higher sensitivity, while a thicker cantilever is desired to increase the critical speed and reduce the wind-induced vibration effect.

In addition, the cantilevers with different Young’s modulus substrates were also tested. PET, PC, and two different PVC films were utilized to fabricate the cantilever with the same size of 30 mm × 15 mm × 0.2 mm. Their responses to different wind speeds are shown in Figure 6. It can be seen that among these materials, the stiffest PVC substrate exhibits the highest critical wind speed. In contrast, the softest PET substrate possesses the lowest critical wind speed and the highest measurement sensitivity. These results are consistent with the qualitative analysis results of Equations (3) and (9).

From the test results above, it can be concluded that a shorter and thicker anemometer substrate with larger stiffness is more conducive to suppressing the wind-induced vibration and increasing the critical wind speed. These results are in concordance with the theoretical analysis results expressed by Equation (9). Based on these results, a microcantilever anemometer with optimized anti-vibration capability (L = 20 mm, W = 15 mm, T = 0.6 mm, E = 3.8 GPa) was fabricated and tested in the tunnel to compare with the anemometer without anti-vibration design (L = 40 mm, W = 15 mm, T = 0.2 mm, E = 2 GPa). The corresponding experimental results are recorded in Figure 7. It can be observed that the anemometer with optimized vibration suppression capability can reach a critical wind speed of 31.10 m/s, achieving an improvement of 697%, compared with 4.46 m/s of the anemometer without anti-vibration design. The increase in the critical wind speed makes the anemometer obtain a wider measurement range for the wind. In contrast, the anemometer without anti-vibration design presents higher sensitivity in low winds. Due to the mutual constraints between measurement range and sensitivity, the performance of the mechanical anemometer needs to be adjusted according to the practical application scenario.

## 4. Conclusions

In this paper, the effect of wind-induced vibration on measurement range of microcantilever anemometers is demonstrated for the first time. The wind-induced vibration theory and corresponding suppression methods for the microcantilever anemometer are developed. Various anemometer prototypes with different structural parameters are fabricated and characterized to examine the effectiveness of the proposed suppression approach. The results indicate that a shorter and thicker cantilever with larger stiffness is expected to suppress the wind-induced vibration and improve the critical wind speed. The suppression of the wind-induced vibration makes the microcantilever anemometer achieve a measurement range improvement of up to 697%. The results presented in this paper can provide a novel idea for widening the measurement range of mechanical anemometers.

## Figures and Tables

**Figure 1 micromachines-13-00720-f001:**
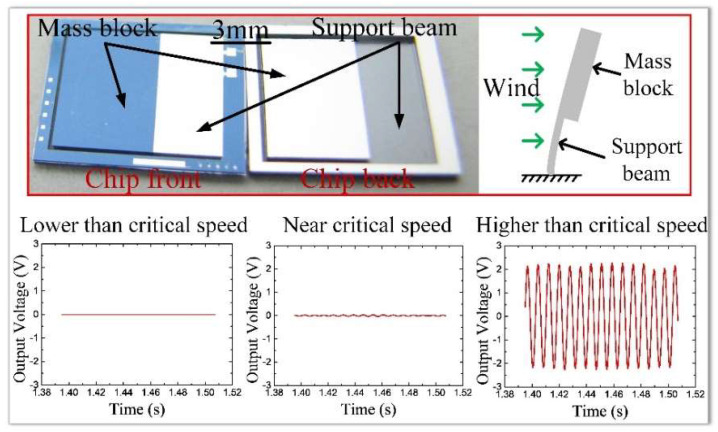
The microcantilever anemometer fabricated by our group, and the effect of wind-induced vibration on the output response of the anemometer under different wind speeds.

**Figure 2 micromachines-13-00720-f002:**
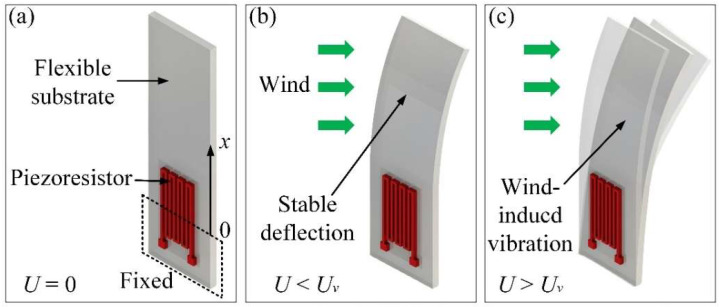
Schematic overview of the microcantilever anemometer under different wind speeds: (**a**) no wind, (**b**) lower than the critical speed, (**c**) higher than the critical speed.

**Figure 3 micromachines-13-00720-f003:**
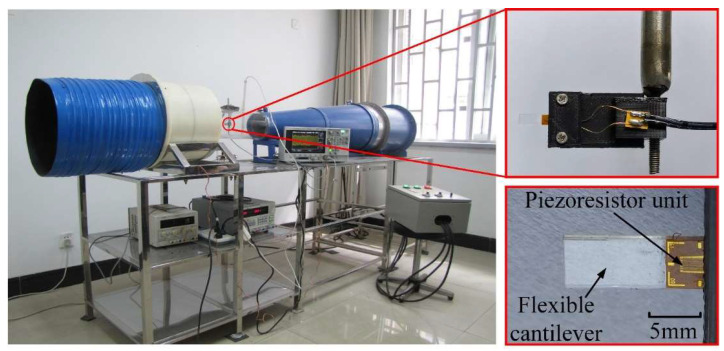
Schematic diagram of the measurement setup and the photograph of the fabricated microcantilever anemometer prototype.

**Figure 4 micromachines-13-00720-f004:**
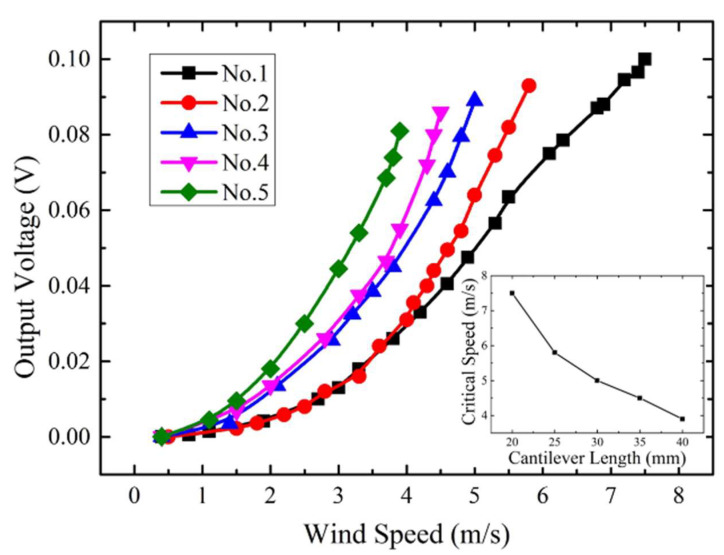
Experimental output voltage versus wind speed characteristics for prototypes with different lengths.

**Figure 5 micromachines-13-00720-f005:**
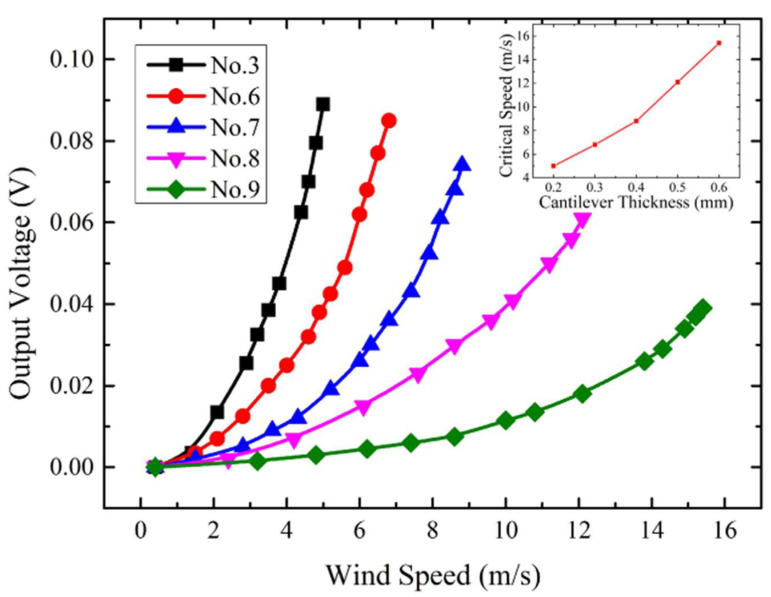
Experimental output voltage versus wind speed characteristics for prototypes with different thicknesses.

**Figure 6 micromachines-13-00720-f006:**
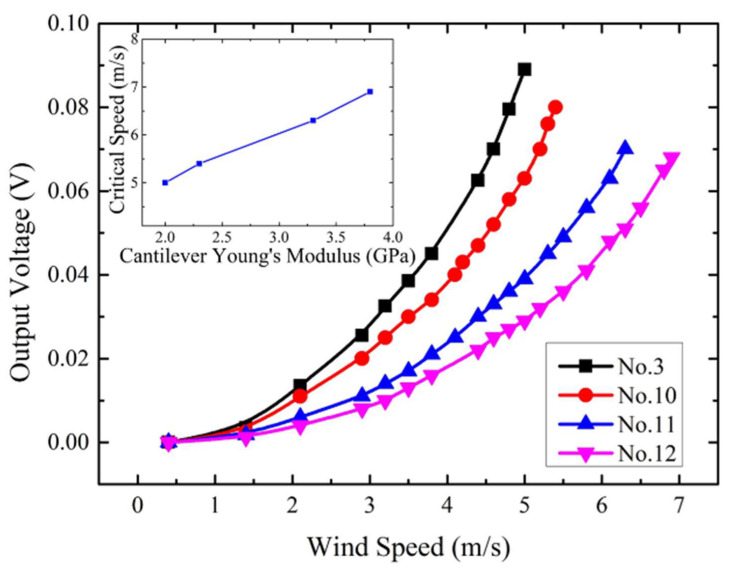
Experimental output voltage versus wind speed characteristics for prototypes with different substrate Young’s modulus.

**Figure 7 micromachines-13-00720-f007:**
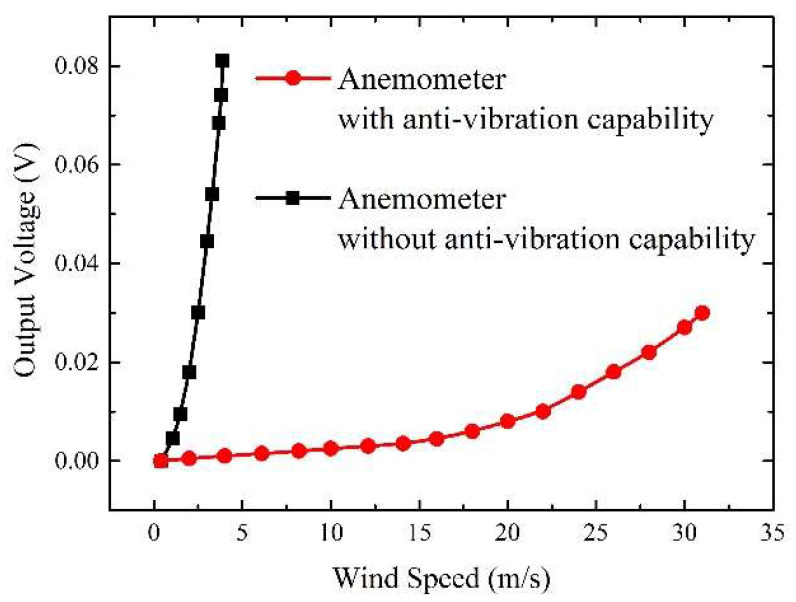
Experimental output voltage versus wind speed characteristics for the anemometers with or without vibration suppression capability.

**Table 1 micromachines-13-00720-t001:** Structural and material parameters of 12 anemometer prototypes.

No.	Length (*L*) (mm)	Width (*W*) (mm)	Thickness (*T*) (mm)	Young’s Modulus (GPa)
1	20	15	0.2	2.0
2	25	15	0.2	2.0
3	30	15	0.2	2.0
4	35	15	0.2	2.0
5	40	15	0.2	2.0
6	30	15	0.3	2.0
7	30	15	0.4	2.0
8	30	15	0.5	2.0
9	30	15	0.6	2.0
10	30	15	0.2	2.3
11	30	15	0.2	3.3
12	30	15	0.2	3.8

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
