# Peer review of "Effect of Wind-Induced Vibration on Measurement Range of Microcantilever Anemometer"

_micromachines, 2022, doi:10.3390/mi13050720_

Round 1

Reviewer 1 Report

This reviewer has found no novelties in the manuscript. And the information are not adequately described (nor to validate the data or reproduce the results).

  • The introduction is poorly written, the references are just thrown there.
  • There is not enough information on how to reproduce the experiment.
  • There are no fluid-dynamics data whatsoever. Even Reynolds numbers?
  • The cantilever beams are not slender, therefore, the approximated vibration equations are not applicable. Or their accuracy is not validated to prove that they are.
  • The conclusions are not giving any new information, i.e. novelties.

Reviewer 2 Report

This paper discusses about the effects of physical structures of MEMS microcantilever anemometers to suppress this vibration. The study aims to reveal the relationship between the wind-induced vibration and the structure of cantilever, theoretically and experimentally

In my opinion, the article is simple and easy to understand. It could be accepted for publication. However, there are some explanations and corrections required to improve the manuscript:

1. Title and Abstract : "this paper aims to reveal the relationship between the substrate wind-induced vibration and its geometries theoretically and experimentally"

This means that the focus of the study should be the effects of the substrate materials and the geometry of the cantilever on the wind induced vibration.

Therefore, in my opinion, the title is not suitable for this study, as the focus is the physical property of the cantilever. My suggestion for the title is : "the effect of the cantilever structures on the critical value of wind induced vibration". 

2. Introduction, LINE 27: it is mentioned that the generated electricity can power wireless micro sensor nodes without a battery.

How could the wind flow provide energy for wireless microsensors ? This statement must be extended with more clear explanation. A schematic could help to explain the potential implementation of the device.

3. It is also required to explain the potential implementation of cantilever based MEMS anemometer at very high wind speed (if working near the critical wind speed).

4. LINE 57: it is mentioned that the vibration behavior of the microcantilever is recorded through the piezoresistor. Unfortunately, there are no informations regarding the specific property of the resistor.

How is the linearity range of the piezoresistor ? How about the sensitivity of the pezoresistor ? How can we determine the critical value of the resistor?

If U<Uv means that the piezoresistor should be working in its linear range, as the W(x) is proportional to resitances. How is the correlation between substrate deformation and the resistance? It is also not clear how to determine that the cantilever deformation is proportional to the output voltage at the range below the critical value.

5. The equation (9) has been the only one justification why the 3 important parameters were chosen that significantly determine the cantilever property 

At what conditions this statement is valid ? how about the damping ? and what other parameters can be ignored ?

6. Table 1 should be extended with substrate material property. Please put a new column for the substrate materials

7. Phrase at LINE 109 and 123 stated that when the fluctuation amplitude of the output voltages exceeds 10 mV, it can be considered that the wind-induced vibration happens

Why the standard decision is the 0.01 V ?. Is it the calibrated value of the piezoresistor ?. Do you have the reference to validate this statement ?

8. LINE 149-152: "Their responses to different wind speeds are shown in Fig. 6. It can be seen that the PVC substrate with the highest Young’s modulus presents the highest critical wind speed. In contrast, the PET substrate with the lowest Young’s modulus possesses the lowest critical wind speed and the highest measurement sensitivity"

What conclusion can be put from these results ?. Is it better to replace the term of young modulus value with more familiar phrase ?, such as flexible, less flexible or very elastic  etc. which could help the reader to understand the property.

9. Phrase at LINE 153 stated that the results are compared with equation (3) and (9).

It is better if the authors put the predicted calculations together with the results in Figure 6, hence we can validate the comparison.

10. LINE 160: two cantilever 160 anemometers A# (L =20 mm, W =15 mm, T =0.6 mm, E =3.8 GPa) and B# (L =40 mm, W =15 161 mm, T =0.2 mm, E =2 GPa).

Are these the parameters considered as extreme parameters (the predicted lowest and highest critical value). Better put specific parameter name for these.

Round 2

Reviewer 1 Report

The authors have revised the manuscript, and answered to the questions. The manuscript is ready for publication.